# The variations of native plasmids greatly affect the cell surface hydrophobicity of sphingomonads

Da Song,[1,2] Xingjuan Chen,[1,2] Hui Yao,[1,2] Guannan Kong,[1,2] Meiying Xu,[1,2] Jun Guo,[1,2] Guoping Sun[1,2]

**ABSTRACT** The organic pollutant-degrading microorganisms with high cell surface hydrophobicity (CSH) are generally favorable due to the positive role of high CSH in pollutant capture and cell colonization. Sphingomonads, an important bacterial group with metabolic versatility, have significant potential for biodegradation and bioremediation of organic pollutants and generally harbor higher CSH than typical Gram-negative bacteria. However, the molecular mechanisms underlying their high CSH are still unclear. In this study, *Sphingobium xenophagum* C1, the most hydrophobic sphingomonad ever known, and its hydrophilic variant C2 were used to identify the genome variations responsible for the CSH difference by comparative genome and transcriptome analysis, as well as gene knockout verification. Our results indicated that the high CSH of strain C1 was largely attributed to the low copy number of the native plasmid p3, which mainly affected the transcriptional levels of outer membrane protein genes through direct and indirect means. In addition, loss of the genes on the native plasmid p5 involved in polysaccharide synthesis and secretion could increase CSH and cell surface friction. The bioinformatics analysis further revealed that some sphingomonad genomes also contained the long homologous fragments and/or key genes of p3 and p5. This study demonstrated the important role of native plasmids in regulating the CSH of sphingomonads, suggesting a novel CSH regulation strategy and evolution process.

**IMPORTANCE** Microbial cell surface hydrophobicity (CSH) reflects nonspecific adhesion ability and affects various physiological processes, such as biofilm formation and pollutant biodegradation. Understanding the regulation mechanisms of CSH will contribute to illuminating microbial adaptation strategies and provide guidance for controlling CSH artificially to benefit humans. Sphingomonads, a common bacterial group with great xenobiotic-degrading ability, generally show higher CSH than typical Gram-negative bacteria, which plays a positive role in organic pollutant capture and cell colonization. This study verified that the variations of two native plasmids involved in synthesizing outer membrane proteins and polysaccharides greatly affected the CSH of sphingomonads. It is feasible to control their CSH by changing the plasmid copy number and sequences. Additionally, considering that plasmids are likely to evolve faster than chromosomes, the CSH of sphingomonads may evolve quickly to respond to environmental changes. Our results provide valuable insights into the CSH regulation and evolution of sphingomonads.

**KEYWORDS** cell surface hydrophobicity, sphingomonads, native plasmids, copy number, outer membrane protein, polysaccharide

Microbial cell surface hydrophobicity (CSH), an important cell surface property, mainly reflects nonspecific adhesion ability and is related to ecological functions and adaptation. Generally, high CSH can enhance adhesion to the surfaces of biological tissues, mineral particles, and artificial materials and increase adsorption capacity

Address correspondence to Meiying Xu, xumy@gdim.cn.

The authors declare no conflict of interest.

See the funding table on p. 15.

for hydrophobic substrates, which therefore promote cell-cell interaction, aggregation, biofilm formation, and substrate utilization (1–3). On the contrary, hydrophilic microbes are likely to disperse and may have high tolerance to hydrophobic chemicals due to little affinity (4, 5). High CSH is usually favorable in organic pollutant biodegradation (6), while low CSH is suitable for liquid-state fermentation (7).

Various cell surface components have been reported to affect microbial CSH, such as surface proteins (8), lipoteichoic acid (9), lipopolysaccharide (LPS) (10), and exopolysaccharide (11). The CSH of the strains from the same species/genus may vary in a large range due to the content differences of certain surface components (12, 13). Moreover, some strains can change their CSH significantly under different culture conditions by regulating surface components (2, 8). Hence, the evolution and regulation of microbial CSH are complex, diverse, and specific. The related knowledge is still limited, especially the underlying molecular mechanisms.

Sphingomonads (the genus *Sphingomonas* in a broader sense including *Sphingobium*) are generally able to degrade a variety of recalcitrant organic compounds, promote plant growth, and assist phytoremediation (14–16). They generally harbor higher CSH than typical Gram-negative bacteria since LPS in the outer membrane is replaced by sphingoglycolipid (SGL), whose carbohydrate part is much shorter and simpler (17, 18). *Sphingobium xenophagum* C1 (formerly *Sphingobium hydrophobicum*), isolated from the electronic waste (e-waste) contaminated sediment, is the most hydrophobic sphingomonad ever known (19). It exhibits great resistance to multiple heavy metals and can degrade common organic pollutants, such as phthalate esters, diphenyl ether, biphenyl, and chlorobenzene (3). Compared with its hydrophilic variant C2 obtained by continuous subculture, it has a higher colonization ability and more adsorption capacity for hydrophobic pollutants (3, 20). The upregulated expression of some outer membrane proteins in strain C1 detected by comparative proteome analysis may be a key hydrophobic factor (20). To further understand the molecular basis underlying the high CSH, we investigated genomic differences between strains C1 and C2 and verified the variations responsible for CSH change. The variations of native plasmids involved in synthesizing outer membrane proteins and polysaccharides were found to affect the CSH significantly, suggesting a novel CSH regulation strategy and evolution process.

## RESULTS

### Genome differences between strains C1 and C2

The complete genome sequences of strains C1 and C2 previously obtained by PacBio RS II sequencing both consisted of two chromosomes (primary chromosome chr1 and secondary chromosome chr2) and five large plasmids (p1, p2, p3, p4, and p5) (3). In order to reveal the genomic variations potentially involved in the change of CSH, the genomes of C1 and C2 were re-sequenced using the Illumina HiSeq X Ten sequencing platform.

The variant calling analysis showed only two single nucleotide polymorphisms (SNPs) and one insertion/deletion (InDel) between the genomes of C1 and C2 (Table 1). The two SNPs led to nonsynonymous mutants of the diguanylate cyclase phosphodiesterase gene (*chr1_233*) and glycoside hydrolase gene (*chr1_1598*), respectively. However, the two sites of C2 were the same as closely related sequences in the Nr database (Fig. S1), suggesting that the stored and sequenced C1 might be a variant of the original C1 rather than the ancestor of C2. The one InDel led to a frameshift mutant of a chromate transporter gene (*chr1_1076*) (Fig. S1). In addition, the transcriptome analysis showed

**TABLE 1** SNPs and InDels between the genomes of strains C1 and C2

| Position | Ref (C1) | Alt (C2) | Influence |
|---|---|---|---|
| chr1: 237,882 | A | G | Diguanylate cyclase phosphodiesterase, L475P |
| chr1: 1,095,320 | TT | –[a] | Chromate transporter, frameshift mutation (247) |
| chr1: 1,642,021 | T | C | Glycoside hydrolase family 3 protein, Y519H |

[a] "–" represents nucleotide deletion on the corresponding position.

that the expression levels of *chr1_233* in the samples of strains C1 and C2 were moderate (fragments per kilobase of transcript per million mapped reads, fpkm > 66), while those of the other two genes were very low (fpkm < 6) (21). The introduction of *chr1_233* from C1 to C2 did not increase the CSH of C2 (Fig. 1a, C2_*chr1_233* 16.1% ± 2.4% vs C2 17.6% ± 2.1%).

The genomic comparison and coverage depth analysis (Fig. 2; Fig. S2) verified that an approximately 100 kb fragment flanked by insertion sequences on p1 was duplicated and translocated into p3 in the sequenced C1 sample (the same sample used for the previous PacBio RS II sequencing). However, the 100 kb fragment translocation did not happen in a new sequenced C1 sample (C1n), i.e., this long fragment was only located on p1, suggesting that its duplication and translocation were probably occasional and unstable. By default, the p3 mentioned below does not include the 100 kb fragment. In addition, an approximately 8 kb fragment flanked by insertion sequences on p3 was lost in the C1 genome (Fig. 2; Fig. S2a). The 8 kb fragment could not also be detected by PCR with specific primers in some offsprings of C2, suggesting its instability. According to the

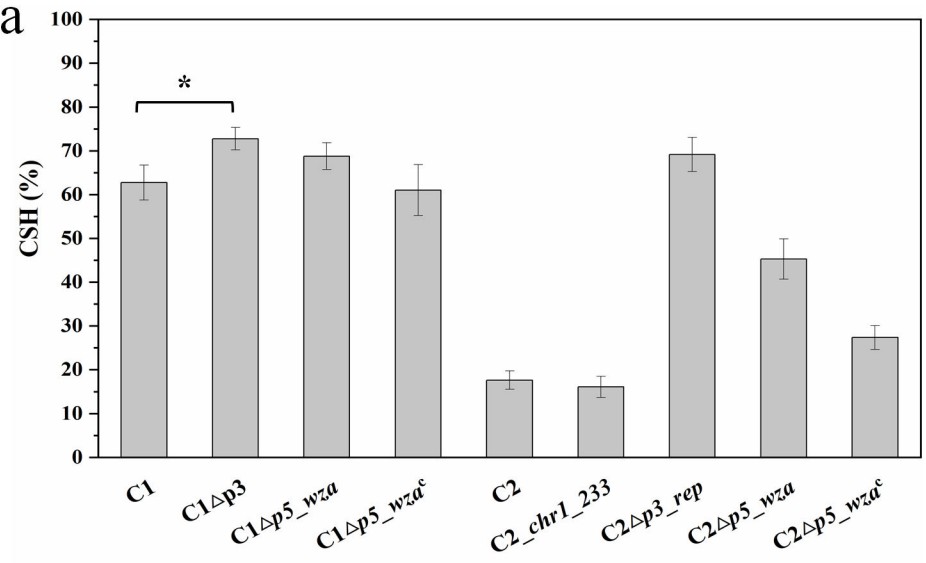

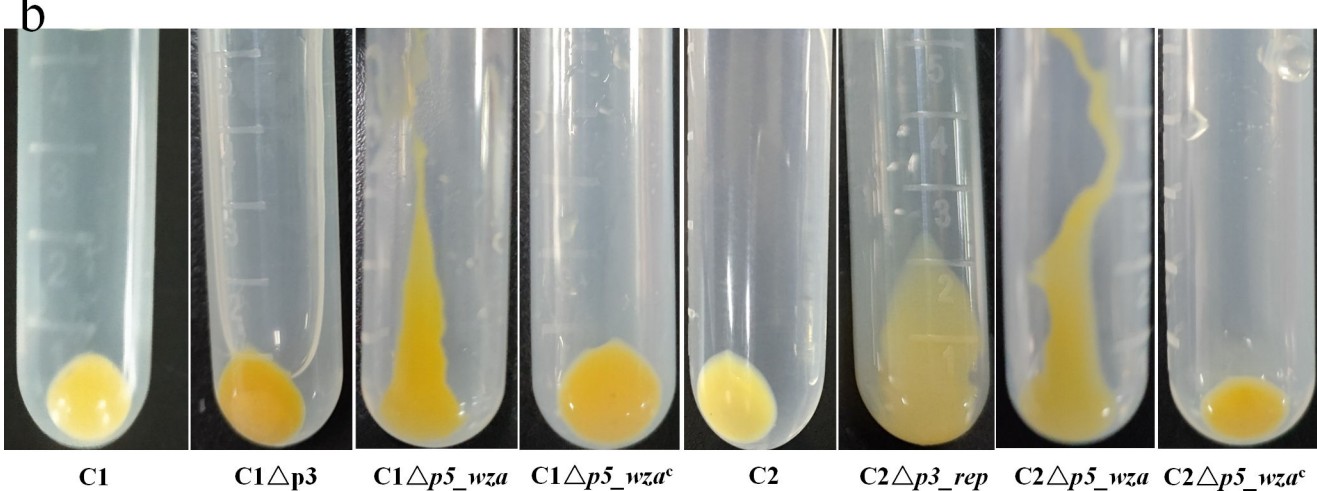

FIG 1 The cell surface features of wild strain and mutants. (a) The CSH determined by the MATH method. According to the computational formula, when all cells in the aqueous phase (PBS buffer) are adsorbed to the organic phase (hexadecane), theoretically, the CSH value is 100%. The values are means of triplicate investigations, and error bars indicate standard deviations. Significant differences were tested by independent-sample *t*-test, *$P < 0.05$. (b) The cell pellet patterns after centrifugation.

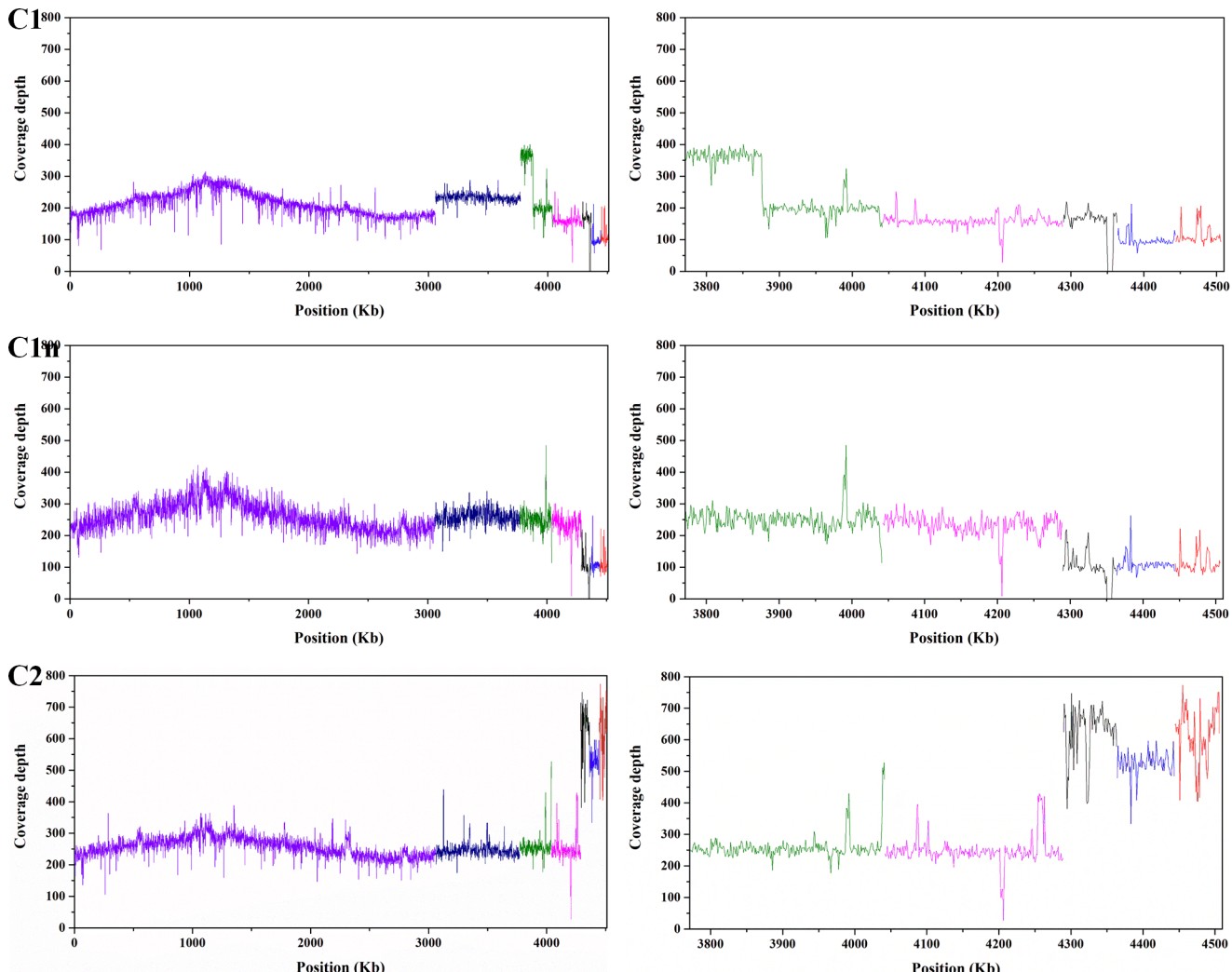

**FIG 2** Sequencing coverage depth of the genomes of strains C1 and C2. In consideration of the translocation fragment on p3 in the C1 genome, the C2 genome is set as a reference. X-axis: the coordinate of the C2 genome; Y-axis: the read number mapped in each position. Each color represents a chromosome or plasmid. The purple, dark blue, green, pink, black, blue, and red correspond to chr1, chr2, p1, p2, p3, p4, and p5, respectively. The figures on the right side are an enlarged display for the coverage depth of five plasmids.

coverage depth, the copy numbers of the three smallest plasmids (p3, p4, and p5) in the C2 genome were 2–4 times more than those in the C1 genome (Fig. 2; Table 2). Moreover, the transcriptome analysis showed that most of the genes on p3 of C2 were significantly upregulated compared with C1 (21).

## Plasmid variations in mutants C2△p3_rep and C1△p3

The plasmid p3 (74.5 kb; excluding the approximately 100 kb fragment translocated from p1) was predicted to have 81 open reading frames (ORFs), encoding three transcriptional regulators (including a TetR family repressor), a TonB-dependent receptor, a set of type IV secretion system, six metabolism enzymes (including two nucleotide sugar synthesis enzymes), and many hypothetical proteins (more than 50% ORFs). In order to verify the relationship between p3 and CSH, the possibly related genes and putative replication initiation protein gene (*rep*) on p3 were tried to be deleted. Due to difficulty in the gene knockout for *Sphingobium xenophagum* strains, especially for plasmid-encoded genes, only the putative *rep* gene on p3 was deleted successfully in strain C2 (C2△p3_rep).

**TABLE 2** Plasmid copy numbers in the genomes of strains C1 and C2 determined by sequencing reads mapping[a]

|  | C1 | C1n | C2 |
|---|---|---|---|
| chr2 | 1.10 | 0.99 | 0.96 |
| p1 | 1.24 | 0.96 | 1.01 |
| p2 | 0.75 | 0.90 | 0.97 |
| p3 | 0.81 | 0.43 | 2.44 |
| p4 | 0.46 | 0.42 | 2.05 |
| p5 | 0.54 | 0.43 | 2.40 |

[a]In consideration of the translocation fragment on p3 in the C1 genome, the C2 genome is set as a reference.

Surprisingly, the deletion of *p3_rep* only led to the reduction in the p3 copy number rather than elimination. The p3 copy number of the mutant C2△*p3_rep* was 0.5–1 determined by real-time quantitative PCR (qPCR), which was similar to strain C1 and about a quarter of strain C2 (Fig. 3a). In addition, approximately 30% of sequences of p3 and p5 were lost compared with C2 (Fig. 3b), which contained the above-mentioned 8 kb fragment on p3 and some polysaccharide synthesis genes on p5. In addition, strain C1 without p3 (C1△p3; Fig. 3b) was occasionally isolated in laboratory cultivation and preservation processes.

## Cell surface characteristics of C2△*p3_rep* and C1△p3

As expected, C2△*p3_rep*, similar to C1, showed much higher CSH than C2 (Fig. 1a, C2△*p3_rep* 69.2% ± 3.9% vs C1 62.8% ± 4.0% vs C2 17.6% ± 2.1%). Moreover, C1△p3 was the most hydrophobic strain (72.8% ± 2.5%), and sometimes the CSH of C1△p3 measured under the same condition was even more than 80%. There was no significant difference in the growth rates of these strains in the Luria-Bertani (LB) liquid medium under aerobic conditions. Interestingly, under different culture media (LB medium and mineral salt medium), different growth stages (exponential phase and early stationary phase), and different centrifuge conditions (6,000 rpm for 5 min and 10,000 rpm for 1 min), mutant C2△*p3_rep* all had a long cell pellet pattern after centrifugation, suggesting an increase in cell surface friction (Fig. 1b). This phenomenon could be related to the loss of polysaccharide synthesis genes on p5. No significant difference in cell surfaces among strains C1, C2, and C2△*p3_rep* was observed using the field emission scanning electron microscope (Fig. S3). In addition, the statistical analysis of cell sizes based on transmission electron microscope images showed that C2 and C2△*p3_rep* were similar and bigger than C1 (Fig. S4).

## Transcriptomic profiles of C1, C2, and C2△*p3_rep*

The transcriptomes of C1 and C2 had been sequenced previously (21). Among the 155 differentially expressed genes (DEGs) between C1 and C2, 35 genes were upregulated, while 120 genes were downregulated in C1. In total, 62 of the DEGs were located on p3, all of which were downregulated. In this study, the transcriptomes of C2 and C2△*p3_rep* cultivated under the same condition were sequenced. The new samples of C2 were denoted as C2n to distinguish them from the previous batch. After removing batch effects, the overall gene transcriptional levels of C1, C2, C2n, and C2△*p3_rep* samples were clustered. It was found that the transcriptomic profiles of the C2△*p3_rep* samples were more similar to those of the C1 samples (Fig. S5). There were 513 DEGs between C2△*p3_rep* and C2n, with 112 upregulated and 401 downregulated in C2△*p3_rep*. Similar to strain C1, the transcriptional levels of most genes on p3 of C2△*p3_rep* were lower compared with C2 although the transcriptional differences of many genes did not meet the DEG criteria (fold change ≥ 2 and false discovery rate < 0.05). Among the 20 DEGs on p3, 18 were downregulated in C2△*p3_rep*, except for a hypothetical protein gene located downstream of *p3_rep* and an integrase gene. In addition, the expression of missing genes on the p3 and p5 of C2△*p3_rep* was not detected.

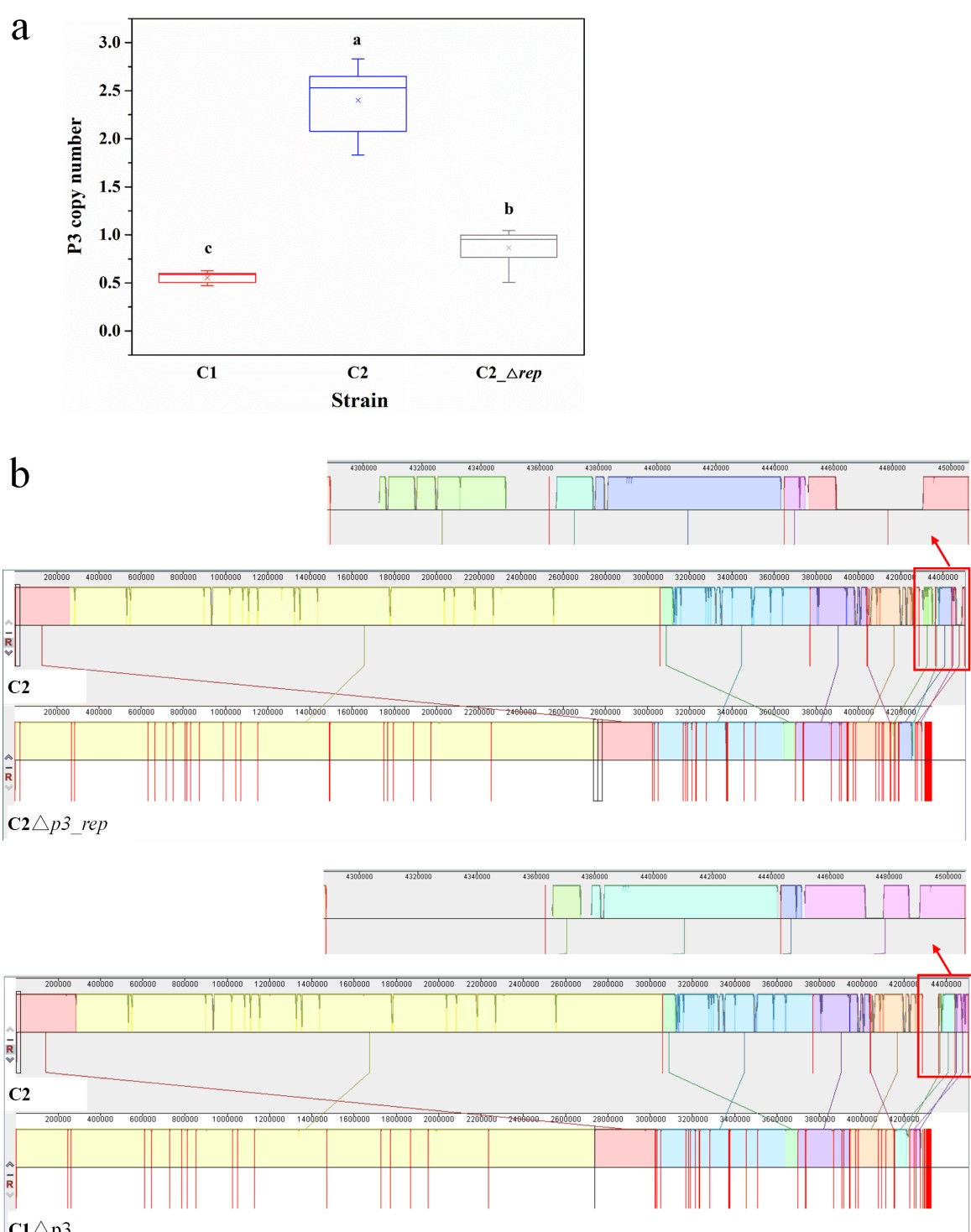

**FIG 3** Plasmid variations in mutants C2△p3_rep and C1△p3. (a) The p3 copy numbers in the genomes of strains C1, C2, and C2△p3_rep identified by qPCR. Significant differences were tested by one-way ANOVA with Bonferroni test ($n = 9$, $\alpha = 0.05$). The a, b, and c represent the differences among these strains. (b) Genome alignment of strains C2, C2△p3_rep, and C1△p3 by Mauve. Each of the colored blocks represents a presumably homologous region. The height of the similarity profile inside each block corresponds to the level of sequence conservation. The regions in the red frames are the three smallest plasmids (P3, P4, and P5).

Considering the DEGs between C1 and C2, as well as the DEGs between C2△p3_rep and C2n, 15 genes were found to be upregulated significantly in both C1 and C2△p3_rep

compared with strain C2. They were all on the p4 and mainly involved the IV secretion system. There were 20 genes downregulated significantly in both C1 and C2△p3_rep compared with strain C2. Among them, 16 were located on p3 and mainly encoded metabolism enzymes, a TonB-dependent receptor protein, a Tetr protein, and hypothetical proteins. The other four genes encoded a pilus protein, two conjugation transfer-related proteins, and a hypothetical protein. Furthermore, the transcriptional levels of all predicted outer membrane protein genes (108 genes) in C1, C2, C2n, and C2△p3_rep samples were analyzed. Although there were a few outer membrane protein genes, which were differentially expressed between C1 and C2 or C2△p3_rep and C2n, the transcriptomic profiles of outer membrane protein genes of C2△p3_rep were also more similar to those of C1 (Fig. 4).

In addition, the enrichment analysis of the KEGG (Kyoto Encyclopedia of Genes and Genomes) pathway showed that only "bacterial secretion system" was significantly enriched between C1 and C2, and that "bacterial secretion system" and "polyketide sugar unit biosynthesis" were significantly enriched between C2△p3_rep and C2n ($q$ value < 0.05) (Table S1). The enrichment of "bacterial secretion system" was attributed to the differential expression of type IV secretion system genes on p3 and p4, as well as protein translocation and secretion genes on the chr1. The enrichment of "polyketide sugar unit biosynthesis" was due to the missing *rfbCBDA* genes on the p5.

## Role of the polysaccharide synthesis genes on P5

The plasmid p5 (62.7 kb) was predicted to have 46 ORFs, approximately 50% of which were involved in the polysaccharide synthesis and secretion. In order to understand their role on CSH and friction, the putative polysaccharide biosynthesis/export protein gene, *wza,* located in the first position of the operon *wza_rfbCBDA* (Fig. S6) on p5 was deleted and complemented in both C1 and C2.

The CSH of C1△p5_wza (68.8% ± 3.1%) was slightly higher than that of C1, while the CSH of C2△p5_wza (45.3% ± 4.6%) was much higher than that of C2 but lower than that of C1 and C2△p3_rep (Fig. 1a). Compared with these *p5_wza* deletion mutants, the complement strains C1△p5_wza<sup>c</sup> (61.0% ± 5.8%) and C2△p5_wza<sup>c</sup> (27.4% ± 2.7%) restored less hydrophobicity (Fig. 1a). These results indicated that the knockout of gene *wza* could lead to a limited increase in CSH. In addition, both C1△p5_wza and C2△p5_wza had a long cell pellet pattern after centrifugation as the same as C2△p3_rep cells, while C1△p5_wza<sup>c</sup> and C2△p5_wza<sup>c</sup> restored like C1 and C2 (Fig. 1b). Furthermore, no matter for wild strains or mutants, neither the removal of extracellular polymeric substances (EPS) by $Na_2$-EDTA and heat treatment nor the digestion of outer membrane proteins by papain and trypsin could cause or change the long cell pellet pattern after centrifugation, indicating that neither EPS nor outer membrane proteins contributed to this phenomenon. With respect to the cell sizes, strain C2 and their mutants (C2△p3_rep, C2△p5_wza, and C2△p5_wza<sup>c</sup>) were similar and bigger than those of strains C1 and C1△p3, indicating that the cell sizes were not associated with CSH (Fig. S4).

## Distribution of the plasmids p3 and p5

According to the Blastn result against the NCBI Database, the long homologous DNA fragments of p3 with coverage ≥10% mainly existed in a few sphingomonads. Among the 109 available complete genomes of sphingomonads (including strain C1) (22), three primary chromosomes, one secondary chromosome, and 19 plasmids were found to contain the long homologous fragments. The plasmid pLA2 of *Novosphingobium pentaromativorans* US6-1 shared the longest homologous fragments (coverage 52%). There was no long homologous fragment in other sequenced *Sphingobium xenophagum* strains PH3-15, QYY, and NBRC 107872. Furthermore, 10 sphingomonads were found to have over 50% genes of p3 by using Blastp with identity ≥50% and coverage ≥50%. In addition, most of the genes on p3 each existed in more than five sphingomonads, and generally, these genes appeared more frequently on plasmids.

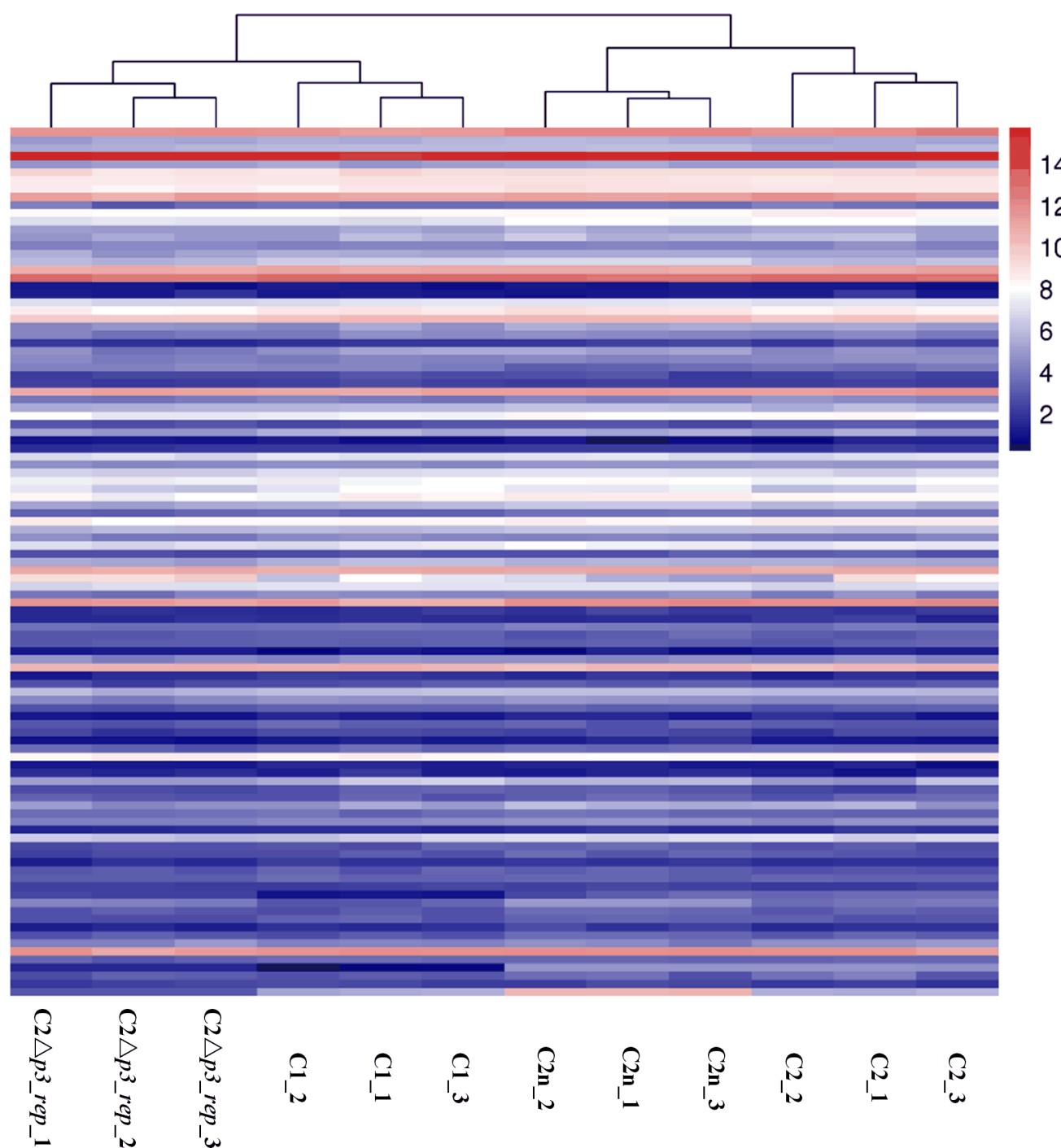

**FIG 4** The transcriptomic profiles of all predicted outer membrane protein genes in C1, C2, C2n, and C2△*p3_rep*. After removing batch effects, the transcriptional levels of all predicted outer membrane protein genes in C1, C2, C2n, and C2△*p3_rep* were used to draw a heatmap.

Similar to the distribution of p3, the long homologous fragments of p5 with coverage ≥10% only existed in a few sphingomonads. There were three primary chromosomes, one secondary chromosome, and seven plasmids from sphingomonads containing the long homologous fragments. The plasmid pSH3 of *Sphingobium xenophagum* PH3-15 shared the longest homologous fragments (coverage 38%). Furthermore, most of the polysaccharide synthesis genes on p5 were found in a part of

the primary chromosomes and plasmids from sphingomonads (Table S2), and generally, these genes appeared more frequently on primary chromosomes.

In addition, the plasmids p3 and p5 both harbored several insert sequences and encoded some transposases, representing a high plasticity. Sometimes, the related sequence variations of p3 and p5 could be observed during the cultivation and preservation processes as shown in the genomes of C1 and C2△p3_rep.

## DISCUSSION

Microbial CSH directly or indirectly affects various physiological characteristics (1). Especially for the organic pollutant-degrading microorganisms, the high CSH can generally promote their colonization ability and degradation efficiency (6). Understanding the molecular mechanisms underlying CSH will contribute to providing guidance for regulating CSH artificially to benefit humans. In this study, we focused on the CSH of sphingomonads, a bacterial group with great xenobiotic-degrading ability, and found that the CSH was greatly affected by the copy number and fragment variations of native plasmids p3 and p5 involved in synthesizing outer membrane proteins and polysaccharides.

It was discovered early that the native plasmids could affect microbial CSH (23, 24). However, there are few research reports on this topic, and the plasmids are diverse and specific, as well as the underlying molecular mechanisms remain unclear. The CSH is affected by various cell surface components and depends on the proportion of hydrophobic components. The plasmids p3 and p5 encode some polysaccharide synthesis enzymes, metabolic enzymes, outer membrane proteins (e.g., TonB-dependent receptor, type IV secretion system, transporter), transcriptional regulatory proteins (e.g., TetR family repressor), and many hypothetical proteins, which may directly or indirectly affect CSH. Generally, polysaccharides are hydrophilic due to multiple hydroxyl groups, and CSH is negatively correlated with exopolysaccharide content (11, 25). The influences of different outer membrane proteins on CSH are varied, and are determined by the proportion of hydrophobic amino acid residues, conformation, and expression quantity (20). The TetR family repressor can control the expression of some outer membrane proteins including efflux pumps (26). There is a crosstalk regulatory network between microbial chromosomes and plasmids. Various plasmid-encoded regulators have been found to regulate the chromosomal genes involved in adherence, biofilm formation, motility, and others (27). The plasmid carriage, e.g., the plasmid pCAR1, can actively impact host phenotypes and transcriptomes (28). In this study, the comparative genome analysis, transcriptome analysis, and gene knockout verification showed that the plasmids p3 and p5 decreased CSH by directly participating in the synthesis of outer membrane proteins and polysaccharides, as well as regulating global metabolism and the expression of cell surface components-related genes on other replicons (Fig. 5).

It is surprising that the deletion of p3_rep reduced the p3 copy number rather than completely eliminating p3. There could be two reasons for the maintenance of p3 after losing the gene p3_rep. First, the other proteins may partly compensate for the role of p3_rep. Only one putative rep gene (p3_rep) was found on p3 by annotation with multiple databases. The encoded Rep protein was similar to two proteins encoded by p4 (coverage 64.6%, identity 31.0%) and p5 (coverage 80.0%, identity 28.8%), respectively, possibly suggesting a functional compensation. Second, the gene annotated as p3_rep is not a real replication initiation protein, but it regulates the replication. Although the p3_rep gene was supported by annotation with the Nr database, protein family domain search, and phylogenetic analysis (22), the highly similar proteins have not been verified experimentally. The most similar Rep protein, which has been verified, is from the plasmid pTAR of *Agrobacterium tumefaciens* (coverage 28.8%, identity 31.9%) (29). In addition, there may be unknown plasmid replication initiation mechanisms. Among the 202 plasmids from sphingomonads available in the NCBI Genome Database, the *rep* genes were not found in 11 plasmids (22).

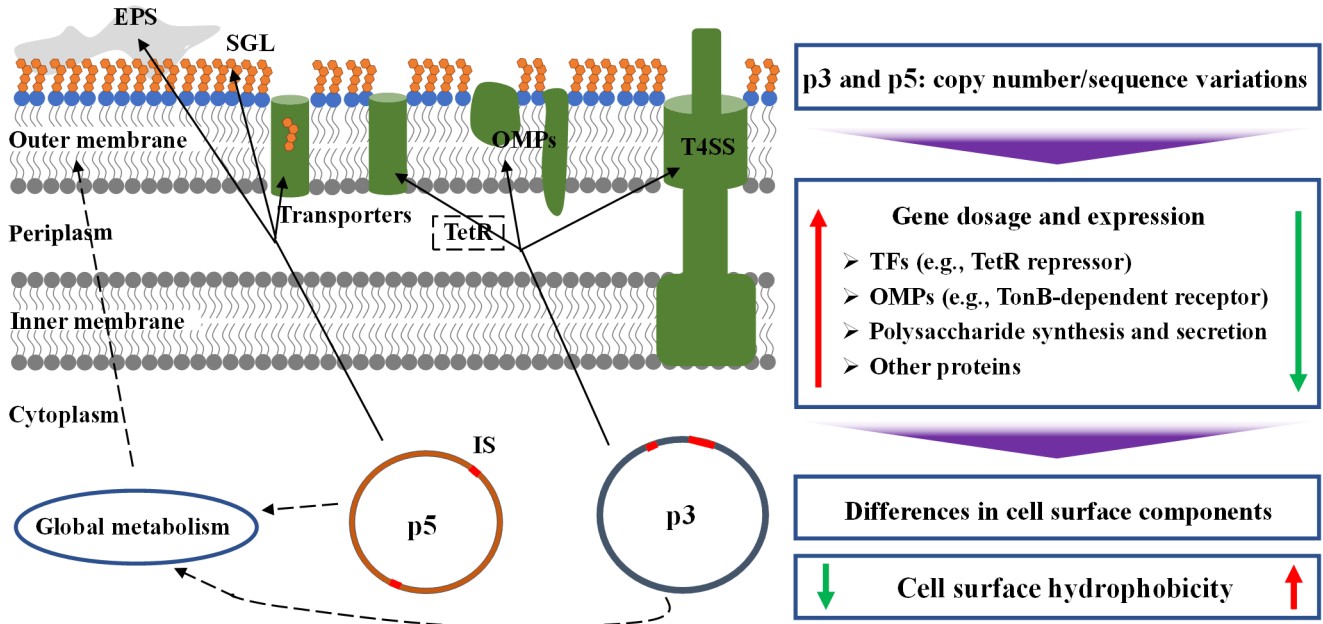

**FIG 5** Proposed influence mechanisms of the native plasmids p3 and p5 on CSH. Solid lines, direct syntheses of cell surface components; dotted lines, indirect effects on cell surface components; EPS, extracellular polymeric substances; SGL, sphingoglycolipid; TFs, transcription factors; OMPs, outer membrane proteins; T4SS, type IV secretion system; IS, insertion sequence.

The long cell pattern pellet after centrifugation reflects high cell surface friction. A similar phenomenon has been observed in the LPS-deficient mutants of *Bradyrhizobium japonicum,* which lack O-antigen composition (a repetitive glycan polymer) and have higher CSH than the wild strain (10). Our study verified that the absence of polysaccharide synthesis genes on the plasmid p5 gave rise to CSH increase and the long cell pellet pattern after centrifugation. The outer membrane of sphingomonads contains SGL instead of LPS (17). Moreover, in this study, the removal of EPS and outer membrane proteins in the wild strain and mutants did not cause or change the long cell pellet pattern. Therefore, this phenomenon could be attributed to the deficiency in the synthesis of carbohydrate parts of tightly bound SGL or other glycolipids. Their compositions, structures, and physiological functions remain to be studied.

Although plasmids generally contain genes involved in environmental adaptation and improve host fitness, plasmid replication and the expression of plasmid-encoded genes can cause metabolic burden (30). The plasmid copy number determines the dosage of these genes and, therefore, influences their expression level (31). The plasmid copy number is regulated appropriately to balance the advantages and costs of plasmid carriage (32, 33). Additionally, the mean of the plasmid copy number is constant at the population level, while the plasmid copy number and expression level of plasmid-encoded genes differ widely among individual cells, which will provide different phenotypes for environmental selection (34, 35). Strain C1 was isolated from the e-waste-contaminated river sediment with an oligotrophic medium containing BDE-209 (19). The low copies of p3, p4, and p5 in the C1 genome may provide enough dosage of beneficial genes at a low cost. Meanwhile, the high CSH promotes cell aggregation, biofilm formation, and colonization, improves stress resistance, and enhances substrate adsorption and uptake (3, 20). The hydrophilic variant C2 was obtained by cell passage in a nutrition-rich LB medium, in which the metabolic burden caused by multicopy plasmids may be negligible. The hydrophilic characteristic enables cells to disperse and be exposed to surrounding nutrients and oxygen. However, the molecular mechanism accounting for the change of plasmid copy number in strains C1 and C2 remains to be studied.

Compared with chromosomes, plasmids and plasmid-encoded traits are likely to evolve faster due to gene dispensability, copy number effect, horizontal gene transfer, frequent recombination, and others (31, 36). Approximately, two-thirds of the available complete genomes of sphingomonads contain 1–11 plasmids with the low copy numbers varying from approximately 0.1 to 7.1 (22). The copy number variation, conjugational transfer, and sequence recombination of plasmids have often been observed among sphingomonads (22, 37, 38). The long homologous DNA fragments and/or key genes of p3 and p5 were also found in some other sequenced chromosomes and plasmids of sphingomonads. The insert sequences and transposases of p3 and p5 can increase their plasticity, and indeed, the related sequence variations have been observed during the host cultivation and preservation processes. In addition, many plasmids encode the genes involved in synthesizing cell surface components. Among 202 plasmids from different sphingomonads (22), 159 plasmids have the genes belonging to the COG category "M, cell wall/membrane/envelope biogenesis," of which 40 plasmids have at least 10 related genes. These plasmids may participate in the CSH regulation of sphingomonads and speed up the CSH evolution.

In summary, the native plasmids p3 and p5 of *Sphingobium xenophagum* C1 involved in the synthesis of outer membrane proteins and polysaccharides can significantly decrease CSH, and the variations of their copy number and sequences can greatly affect CSH. Some sphingomonads also have long homologous fragments and key genes of these two plasmids, which are likely to affect their CSH. Future studies on other native plasmids related to cell surface components' synthesis, plasmid-encoded hypothetical proteins, plasmid influences on global metabolism, plasmid variation mechanisms, and cell surface components will further illuminate the roles of native plasmids in CSH regulation and evolution of sphingomonads.

## MATERIALS AND METHODS

### Bacterial strains, plasmids, and growth conditions

The bacterial strains and plasmids used in this study are described in Table 3. *Sphingobium xenophagum* C1 (= CCTCC AB 2015198) was isolated from e-waste-contaminated sediment in Guiyu, China, with a mineral medium containing deca-brominated diphenyl ether (BDE-209) (19). Strain C2 (= CCTCC AB 2015427), a hydrophilic variant of C1, was obtained by cell passage for ~100 generations in the nutrition-rich LB medium (20). Mutant C1△p3 lacking the native plasmid p3 was also isolated from the subculture of C1. *Escherichia coli* and *S. xenophagum* strains were cultured aerobically (180 rpm for tubes and shakers) in LB medium at 37°C and 30°C, respectively. Kanamycin, streptomycin, and gentamycin were added when needed.

### Genome sequencing and analysis

The complete genomes of strains C1 and C2 had been assembled by SMRT analysis pipeline with PacBio RS II sequencing reads (3). They were re-sequenced using the Illumina HiSeq X Ten sequencing platform to polish the previous sequences. The genomes of mutants C1△p3 and C2△*p3_rep* were also sequenced on the Illumina HiSeq X Ten sequencing platform.

#### *Genome sequence correction*

The raw sequencing data were filtered using fastp (41) and then mapped to the previously assembled genome sequences by BWA-MEM (42). The mapped reads were used to polish the genome sequences by Pilon (43). ORFs were re-predicted and annotated by Prokka, BlastKOALA, and EggNOG-mapper (44–46). The subcellular locations of encoded proteins were predicated by PSORTb 3.0 (47).

**TABLE 3** Strains and plasmids used in this study

| Strain/plasmid | Description | Reference or source |
|---|---|---|
| *S. xenophagum* strains | | |
| C1 | Wild-type, hydrophobic strain | Lab stock |
| C1△p3 | The variant of C1 lacking P3 | This study |
| C1△*p5_wza* | The *p5_wza* deletion mutant derived from C1 | This study |
| C1△*p5_wza*[c] | The complement strain of C1△*p5_wza* | This study |
| C2 | The hydrophilic variant of C1 | Lab stock |
| C2△*p3_rep* | The *p3_rep* deletion mutant derived from C2 | This study |
| C2△*p5_wza* | The *p5_wza* deletion mutant derived from C2 | This study |
| C2△*p5_wza*[c] | The complement strain of C2△*p5_wza* | This study |
| C2_*chr1_233* | C2 with pBBR1MCS-5 containing the gene *chr1_233* from C1 | This study |
| *E. coli* strains | | |
| DH5α | Host for pAK405 and pBBR1MCS-5 derivatives | TransGen Biotech Co., Ltd (China) |
| Trans1-T1 | Host for pEASY-T1 derivatives | TransGen Biotech Co., Ltd (China) |
| Plasmids | | |
| pBBR1MCS-5 | Broad-host vector; Gm[r] | (39) |
| pAK405 | Suicide vector for *Sphingobium*; Km[r], Sm[s] | (40) |
| pEASY-T1 (simple) | Cloning vectors; Km[r] | TransGen Biotech Co., Ltd (China) |

## Variant calling

The quality trimmed reads of strain C2 were mapped to the polished genome sequences of strain C1 by BWA-MEM. The mapping data were then used to detect SNPs and InDels between the strains according to the Genome Analysis Toolkit Best Practices Pipeline (39). In addition, the genomes of mutants C1△p3 and C2△*p3_rep* were assembled by SPAdes (40) and were aligned with the C1 and C2 genome sequences using Mauve (48). The genomic differences were further confirmed by PCR and Sanger sequencing of the relevant loci.

## Plasmid copy number determination by sequencing reads mapping

The mapping data based on sequencing reads from the Illumina platform were used to calculate the coverage depth of each base by Samtools (49). The bases covered with no read were ignored. The plasmid copy number was calculated as the ratio of the average coverage depth of the plasmid to the chromosome.

## Gene deletion

The in-frame deletion mutants were constructed through two-step homologous recombination as described previously with slight modifications (50). The in-frame deletion design would avoid affecting the transcription of the downstream genes. Briefly, 600–900 bp of the upstream and downstream regions flanking the target gene were joined by overlap extension PCR and then ligated into pAK405 and cloned in DH5α. The pAK405 derivative was extracted and transformed into a wild strain by electroporation (Eppendorf Multiporator; 2.1 kV, 5 ms). Different from the previous method, Kanamycin (20 µg/mL) and streptomycin (120 µg/mL) were used in the subsequent two-step selection processes, respectively, and the recombination events were verified by PCR with specific primers. Finally, the deletion mutant was confirmed by Sanger sequencing of the mutated region.

## Gene complementation

The broad-host vector pBBR1MCS-5 was used in gene complementation (51). For the expression of the gene *chr1_233* (diguanylate cyclase phosphodiesterase gene) of strain C1 in strain C2, a fragment containing this gene and its upstream promoter was PCR amplified and then ligated into pBBR1MCS-5 and cloned in DH5α. The pBBR1MCS-5 derivative was extracted and transformed into C2 by electroporation (Eppendorf Multiporator; 2.1 kV, 5 ms). The cells with pBBR1MCS-5 containing *chr1_233* were selected by gentamycin and confirmed by sequencing the ligated fragment.

Similarly, for the complementation of C1△*p5_wza* and C2△*p5_wza*, a fragment containing the operon *wza_rfbCBDA* (including the promoter) was generated by PCR and then ligated into pBBR1MCS-5 and cloned in DH5α. The pBBR1MCS-5 derivative was transformed into the mutants using the above method.

## Plasmid copy number determination by qPCR

The copy numbers of *chr1_1209* (the *gyrb* gene on the primary chromosome chr1) and *p3_26* (the *virB1* gene on the plasmid p3) in the same genomic DNA sample were quantified absolutely by qPCR. Briefly, the target sequences were PCR amplified and cloned in pEASY-T1 to produce DNA standards. The 10-fold serial dilution series of the DNA standards ranging from $\sim 6 \times 10^2$ to $6 \times 10^7$ copies/μL were used to construct the standard curves. The qPCR amplification and detection were performed using a Bio-Rad CFX Real-time PCR system. The 25 μL of reaction mixture contained 12.5 μL of 2× SYBR Premix Ex Taq II (TaKaRa), 1 μL of each primer (10 μM), 2 μL of DNA template (~10 ng/μL), and 8.5 μL of ddH$_2$O. The amplification procedure was set up as follows: 95℃ for 30 s, followed by 40 cycles of 95℃ for 5 s, 56℃ for 30 s, and 72℃ for 15 s. The cycle threshold values were determined and used to calculate copy numbers of target sequences by the fitted formulas of standard curves. All assays were performed in nonuplicate (three biological repeats per strain and three technical repeats per biological repeat). The p3 copy number was calculated as the copy number ratio of *p3_26* to *chr1_1209*.

## Transcriptome sequencing and analysis

The transcriptomes of strains C1 and C2 cultivated with mineral salt medium had been sequenced (21). In this study, the transcriptomes of strains C2 and C2△*p3_rep* cultivated under the same condition were sequenced and analyzed. The cells of C2 and C2△*p3_rep* were harvested in triplicate. The mRNA of each sample was sequenced using the Illumina HiSeq 2500 sequencing platform. Raw data were filtered by Fastp (41). Quality trimmed reads were mapped to the reference genome using Bowtie2 (52), and reads mapped to ribosome RNA were removed. The gene expression was calculated using RSEM (53). The gene expression level was normalized by using the fragments per kilobase of transcript per million mapped reads method. The edgeR package was used to identify DEGs across samples with fold change ≥2 and false discovery rate <0.05 (54). The DEGs were then subjected to enrichment analysis of KEGG pathways on the Omicshare platform (https://www.omicshare.com/tools/). The batch effects between the two sets of transcriptome data (C1 vs C2 and C2n vs C2△*p3_rep*) were removed using the ComBat function of the R package SVA (55). The heatmap drawn using the Omicshare tools was used to show the transcriptomic profiles of C1, C2, C2n, and C2△*p3_rep* samples.

## CSH assay

The CSH was measured using the microbial adhesion to hydrocarbon (MATH) protocol with slight modifications (20). Briefly, the cells in the late-exponential growth phase were washed three times and resuspended with phosphate-buffered saline (PBS) buffer to an OD$_{600}$ of ~0.6 ($A_0$). And then, 800 μL of hexadecane was added to 4 mL of cell suspension. The mixture was vortexed vigorously for 30 s and then allowed for phase separation for 30 min. The OD$_{600}$ of aqueous phase was remeasured ($A_1$). The CSH value

was calculated using the equation: CSH = $(A_0 - A_1)/A_0$. All assays were performed in triplicate.

## EPS removal

The EPS were extracted by $Na_2$-EDTA or heat treatment as described previously (56, 57). Briefly, the cells in the late-exponential growth phase were collected and washed three times with PBS buffer. The cells were resuspended in PBS buffer containing 2% $Na_2$-EDTA and incubated at 4°C for 3 h, or resuspended in 0.1 M pH 7.5 Tris/HCl solution and treated at 75°C with shaking at 150 rpm for 1 h.

## Protease digestion

The cells in the late-exponential growth phase were collected and washed three times with PBS buffer. The cells were resuspended in an equal volume of PBS buffer containing 150 µg/mL of trypsin or 200 µg/mL papain. The mixtures were incubated at 30°C for 45 min.

## Cell observation by scanning electron microscopy and transmission electron microscope

The cells in the late-exponential growth phase were collected, fixed in 3% glutaraldehyde, washed with PBS, dehydrated by ethanol, exchanged by tertiary butanol, and freeze-dried. The dried samples were coated with gold and imaged using a field emission scanning electron microscope (Zeiss Merlin).

The cell suspension in the late-exponential growth phase was dropped onto the copper mesh and then 3% phosphotungstic acid was added to the copper mesh. The copper mesh was washed with distilled water. The sample on the air-dried mesh was imaged using a transmission electron microscope (Hitachi). All cells in the pictures were measured with ImageJ for cell size comparison (58). For each strain, more than 30 cells were analyzed statistically.

## ACKNOWLEDGMENTS

We thank Professor Jian He from Nanjing Agricultural University for the guidance on the gene knockout experiment.

This work was supported by the National Key Research and Development Program of China (2021YFA0910300), the National Natural Science Foundation of China (32171409), the Science and Technology Project of Guangzhou (202103000086), the Science and Technology Project of Guangdong Province (2022A0505090004), the Guangdong Basic and Applied Basic Research Foundation (2022A1515110687), and GDAS' Project of Science and Technology Development (2022GDASZH-2022010101).

D.S., X.C., G.S., J.G., and M.X. conceived and designed the research. D.S., H.Y., and G.K. performed the experiments. D.S., X.C., and M.X. analyzed the data. D.S. wrote the manuscript. All authors revised and approved the manuscript.

## AUTHOR AFFILIATIONS

[1]Guangdong Provincial Key Laboratory of Microbial Culture Collection and Application, State Key Laboratory of Applied Microbiology Southern China, Institute of Microbiology, Guangdong Academy of Sciences, Guangzhou, Guangdong, China
[2]Guangdong Environmental Protection Key Laboratory for Microbiology and Regional Ecological Safety, Institute of Microbiology, Guangdong Academy of Sciences, Guangzhou, Guangdong, China

## AUTHOR ORCIDs

Da Song (iD) http://orcid.org/0000-0001-8064-3409

Meiying Xu  http://orcid.org/0000-0001-7276-4219

## FUNDING

| Funder | Grant(s) | Author(s) |
|---|---|---|
| MOST \| National Key Research and Development Program of China (NKPs) | 2021YFA0910300 | Meiying Xu |
| MOST \| National Natural Science Foundation of China (NSFC) | 32171409 | Xingjuan Chen |
| Bureau of Science and Information Technology of Guangzhou Municipality \| Guangzhou Municipal Science and Technology Project (Guangzhou Science and Technology Plan) | 202103000086 | Xingjuan Chen |
| GDSTC \| Science and Technology Planning Project of Guangdong Province (S&T Project of Guangdong Province) | 2022A0505090004 | Meiying Xu |
| GDSTC \| Basic and Applied Basic Research Foundation of Guangdong Province (廣東省基礎與應用基礎研究專項資金) | 2022A1515110687 | Guannan Kong |
| GDAS' Project of Science and Technology Development | 2022GDASZH-2022010101 | Meiying Xu |

## DATA AVAILABILITY

The raw sequence data reported in this study have been deposited in the Genome Sequence Archive in BIG Data Center (Beijing Institute of Genomics, Chinese Academy of Sciences) (59) under accession number CRA002338 and the National Omics Data Encyclopedia (Shanghai Institute of Nutrition and Health, Chinese Academy of Sciences) under accession numbers OEP004335 and OEP002610, and can be viewed through the URLs: https://www.biosino.org/node/review/detail/OEV000466?code=A63VJQOG; https://www.biosino.org/node/review/detail/OEV000472?code=ILLUUWFV.

## ADDITIONAL FILES

The following material is available online.

### Supplemental Material

**Fig. S1 (mSystems00862-23_S0001.tif).** Sequence alignments of the mutant proteins.
**Supplemental figure legends (mSystems00862-23_S0002.docx).** Legends for Fig. S1 to S6.
**Fig. S2 (mSystems00862-23_S0003.tif).** Sequence alignments of the p1 and p3 in the C1 and C2 genomes.
**Fig. S3 (mSystems00862-23_S0004.tif).** The field emission scanning electron microscope images.
**Fig. S4 (mSystems00862-23_S0005.tif).** The cell sizes of C1, C2, and their mutants.
**Fig. S5 (mSystems00862-23_S0006.tif).** Hierarchical clustering of the transcriptomic profiles.
**Fig. S6 (mSystems00862-23_S0007.tif).** Genetic organization of the operon wza_rfbCBDA.
**Tables S1 and S2 (mSystems00862-23_S0008.docx).** The enriched KEGG pathways and distributions of polysaccharide synthesis genes.

### Open Peer Review

**PEER REVIEW HISTORY (review-history.pdf).** An accounting of the reviewer comments and feedback.

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
