## [Reviewer comments · mSystems]

The variations of native plasmids greatly affect cell surface hydrophobicity of sphingomonads

Da Song, Xingjuan Chen, Hui Yao, Guannan Kong, Meiyong Xu, Jun Guo, and Guoping Sun

Corresponding Author(s): Meiyong Xu, Institute of Microbiology, Guangdong Academy of Sciences

Review Timeline:

Submission Date:

August 16, 2023

Accepted:

September 26, 2023

Editor: Xiao-Hua Zhang

Reviewer(s): The reviewers have opted to remain anonymous.

Transaction Report:

DOI: <https://doi.org/10.1128/mSystems.00862-23>

September 26, 2023

Prof. Meiyong Xu
Institute of Microbiology, Guangdong Academy of Sciences
100, Central Xianlie Road
Guangzhou, Guangdong 510070
China

Re: mSystems00862-23 (The variations of native plasmids greatly affect cell surface hydrophobicity of sphingomonads)

Dear Prof. Meiyong Xu:

Your manuscript has been accepted, and I am forwarding it to the ASM Journals Department for publication. For your reference, ASM Journals' address is given below. Before it can be scheduled for publication, your manuscript will be checked by the mSystems production staff to make sure that all elements meet the technical requirements for publication. They will contact you if anything needs to be revised before copyediting and production can begin. Otherwise, you will be notified when your proofs are ready to be viewed.

If you would like to submit a potential Featured Image, please email a file and a short legend to msystems@asmusa.org. Please note that we can only consider images that (i) the authors created or own and (ii) have not been previously published. By submitting, you agree that the image can be used under the same terms as the published article. File requirements: square dimensions (4" x 4"), 300 dpi resolution, RGB colorspace, TIF file format.

We recognize that the video files can become quite large, and so to avoid quality loss ASM suggests sending the video file via <https://www.wetransfer.com/>. When you have a final version of the video and the still ready to share, please send it to mSystems staff at msystems@asmusa.org.

Sincerely,

Xiao-Hua Zhang
Editor, mSystems

Journals Department
E-mail: mSystems@asmusa.org